# A Novel Cartesian Plot Analysis for Fixed Monolayers That Relates Cell Phenotype to Transfer of Contents between Fibroblasts and Cancer Cells by Cell-Projection Pumping

**DOI:** 10.3390/ijms23147949

**Published:** 2022-07-19

**Authors:** Swarna Mahadevan, Kenelm Kwong, Mingjie Lu, Elizabeth Kelly, Belal Chami, Yevgeniy Romin, Sho Fujisawa, Katia Manova, Malcolm A. S. Moore, Hans Zoellner

**Affiliations:** 1The Cellular and Molecular Pathology Research Unit, Oral Pathology and Oral Medicine, School of Dentistry, Faculty of Medicine and Health, The University of Sydney, Westmead Hospital, Westmead, NSW 2145, Australia; smah9946@uni.sydney.edu.au (S.M.); kenelm.kwong@health.nsw.gov.au (K.K.); mingjielu@gmail.com (M.L.); elizabeth.kelly@sydney.edu.au (E.K.); belal.chami@sydney.edu.au (B.C.); 2The School of Medical Sciences, The University of Sydney, Sydney, NSW 2006, Australia; 3Molecular Cytology, The Memorial Sloan Kettering Cancer Center, 415-417 E 68 Street, ZRC 1962, New York, NY 10065, USA; rominy@mskcc.org (Y.R.); sfujisawa@hibercell.com (S.F.); k-manova@ski.mskcc.org (K.M.); 4Cell Biology, The Memorial Sloan Kettering Cancer Center, 430 E 67th St, RRL 717, New York, NY 10065, USA; moorem@mskcc.org; 5Biomedical Engineering, Faculty of Engineering, The University of Sydney, Sydney, NSW 2006, Australia; 6Graduate School of Biomedical Engineering, University of NSW, Kensington, NSW 2052, Australia; 7Strongarch Pty Ltd., Pennant Hills, NSW 2120, Australia

**Keywords:** cell-projection pumping, osteosarcoma, cancer, cancer associated fibroblasts, co-culture, Cartesian plot, image analysis, DNA methylation, image analysis, cell morphology

## Abstract

We recently described cell-projection pumping as a mechanism transferring cytoplasm between cells. The uptake of fibroblast cytoplasm by co-cultured SAOS-2 osteosarcoma cells changes SAOS-2 morphology and increases cell migration and proliferation, as seen by single-cell tracking and in FACS separated SAOS-2 from co-cultures. Morphological changes in SAOS-2 seen by single cell tracking are consistent with previous observations in fixed monolayers of SAOS-2 co-cultures. Notably, earlier studies with fixed co-cultures were limited by the absence of a quantitative method for identifying sub-populations of co-cultured cells, or for quantitating transfer relative to control populations of SAOS-2 or fibroblasts cultured alone. We now overcome that limitation by a novel Cartesian plot analysis that identifies individual co-cultured cells as belonging to one of five distinct cell populations, and also gives numerical measure of similarity to control cell populations. We verified the utility of the method by first confirming the previously established relationship between SAOS-2 morphology and uptake of fibroblast contents, and also demonstrated similar effects in other cancer cell lines including from melanomas, and cancers of the ovary and colon. The method was extended to examine global DNA methylation, and while there was no clear effect on SAOS-2 DNA methylation, co-cultured fibroblasts had greatly reduced DNA methylation, similar to cancer associated fibroblasts.

## 1. Introduction

We earlier described the exchange of membrane, organelles and cytoplasmic protein between cultured human fibroblasts and cancer cells [1,2,3,4], and others have made similar observations [5,6,7,8,9,10,11,12,13,14,15,16,17,18,19,20,21,22,23,24,25,26,27,28]. Uptake of cellular contents by such transfers may be via exosomes, tunnelling nanotubes, or a mechanism we recently described and term ‘cell-projection pumping’, and these can significantly change the phenotype of acceptor cells [2,3,5,6,7,8,9,10,11,12,13,14,15,16,17,18,19,20,21,22]. Notably, a major role for either exosomes or tunnelling nanotubes is excluded in our cultures by video microscopy demonstrating highly localized, discrete and rapid transfer events independent of tunnelling nanotubes and inconsistent with exosomes, as well as in experiments with conditioned media containing putative exosomes [1,2,3,4], so that cell-projection pumping is responsible for the transfers we study. Transfer of mitochondria appears especially significant, and can confer chemotherapy resistance to cancer cells [5,7,8,21,22,23,24,25,26,27,28]. We are made doubtful, however, that mitochondria are the only or prime agents mediating phenotypic changes, by our observation that in addition to mitochondria, there is also bulk transfer of: cytoplasmic proteins; plasma membrane bound alkaline phosphatase; and organelles smaller than mitochondria [1,2,3]. Our doubts are further raised by recent work showing that most phenotypic effects in cancer cells of receipt of fibroblast contents, are lost upon cancer cell mitosis. This would not be expected if mitochondria were critical, because these organelles survive mitosis and would be expected to maintain in daughter cells any influence they may have had in the mother [4].

Most of our work has been with co-cultures of human fibroblasts with SAOS-2 osteosarcoma cells, because it has been helpful to build understanding with quantitative studies in a single cancer cell line; thus limiting the variability inherent to study of differing cancer cell types [1,2,3,4,29,30]. Nonetheless, we have also studied transfer between fibroblasts and a wide range of further cancer cell lines, including: osteosarcoma cell lines; melanoma cell lines; ovarian cancer cell lines; colon cancer cell lines; and lung cancer cells [1,2,3]. To examine the effect on cancer cell phenotype, we compared the cell-profile area and cell circularity of the population of SAOS-2 co-cultured with fibroblasts that were dual labelled by uptake of fibroblast fluorescence, with that of SAOS-2 without evident fibroblast fluorescence [1]. We also studied SAOS-2 and other cancer cells co-cultured with fibroblasts and then separated by fluorescence activated cell sorting (FACS) on the basis of the level of fibroblast label uptake [2]; while we further performed single-cell tracking analysis from time-lapse recordings of co-cultures of SAOS-2 with fibroblasts [4]. In those studies, we found that uptake of fibroblast contents by cancer cells increases cancer cell: proliferation; migration; cell-size; and internal cell complexity [1,3,4]. In separate related work, we showed that SAOS-2 altered cytokine synthesis of fibroblasts in a contact dependent manner [30].

Morphological changes in cancer cells are of fundamental interest, because they indicate emergence of cancer cell diversity, which is an important driver for clinical progression and treatment resistance [31]. Morphological changes are further important, because the extent of diversity amongst cancer cells is a diagnostic marker and correlates with poor prognosis in histopathological diagnosis [31]. In addition to the phenotypic effects of uptake of fibroblast contents by cancer cells already mentioned, we have observed shifts in SAOS-2 cell-profile area and cell circularity at the cell-population level, studying fixed monolayers of SAOS-2 co-cultured with fibroblasts [1]. Although the cell-population level analysis earlier possible with fixed monolayers lacked capacity to provide meaningful information at the single cell level [1], it is reassuring that recent single-cell tracking experiments support our earlier description of these morphological effects [4].

The broad range of ‘starting fluorescence’ seen in individual fibroblast and cancer cell control populations cultured alone, poses a special difficulty for interpretation of fluorescence transfer in fixed monolayers. In particular, it has not been possible to infer if any given cell had received a lot of material from an opposing cell type that had low levels of initial fluorescence, or alternatively, if it had received only a small amount of material from an opposing cell that happened to have very high levels of initial fluorescence. This enhanced further ambiguity as to the precise identity of individual dual labelled cells. Despite the power of single-cell tracking [4,32,33], it has become clear that it would be advantageous to have access to a further quantitative method that improves identification of cells and quantitation of fluorescence transfer in fixed monolayers. Such a method would make it possible to better correlate changes in expression of markers detectable by any in situ method, with the extent of transfer between cells. Importantly, such a method would also permit further analysis of phenotypic properties that cannot be examined by single-cell tracking, but which require the use of fixed monolayers for study. DNA methylation is one such property.

DNA methylation is an important epigenetic modification with profound effects on cell phenotype [34], and is appreciably changed in cancer cells [35,36,37], while heterogeneity in DNA methylation contributes to tumor cell diversity [38,39]. From this it is of fundamental interest to know if uptake of fibroblast contents by cell-projection pumping affects cancer cell DNA methylation. 

Cancer associated fibroblasts are an important component of the tumor microenvironment, and influence responsiveness to cancer treatment via a range of mechanisms including their: effect on cancer cell stemness [40]; increasing epithelial to mesenchymal transformation via transforming growth factor-β release [41]; modifying chemotherapy responsiveness [42]; and aiding immune evasion through secretory products and evocation of a dense fibrotic stroma that obstructs lymphocyte migration [43,44,45]. Because of this, cancer associated fibroblasts have been identified as a target for therapeutic intervention [46]. It is of interest for the current work that cancer associated fibroblasts have hypomethylated DNA [27,47,48,49,50]. 

In the current report, we describe a novel Cartesian plot image analysis method that quantitates transfers of fluorescent label between populations of co-cultured cells observed in fixed monolayers. This overcomes some of the limitations of previous approaches, by providing information on sub-populations of co-cultured cells with regard to their precise identity, and the extent of transfer they have experienced relative to control cells cultured alone. We demonstrate utility of this new quantitative method by confirming the earlier reported effect of the uptake of a fibroblast fluorescent marker on SAOS-2 cell circularity and cell-profile area [1,4], and use this approach to further study similar effects across a range of cancer cell lines. We demonstrate wide applicability of the method for analysis of non-morphological features, by studying global DNA methylation in co-cultures of SAOS-2 with human dermal fibroblasts. 

## 2. Results

### 2.1. Fluorescent Labels Transferred between Co-Cultured Cell Populations

Transfer of fluorescent labels consistent with cell-projection pumping was seen across all experiments, with the most evident transfer being from fibroblasts to cancer cells. Figure 1 shows representative co-cultures of fibroblasts with SAOS-2 as well as with Colo316. Appearances were similar across other co-cultures. 

### 2.2. Cartesian Plot Analysis Demonstrated Five Sub-Populations of Cells in Co-Cultures

Figure 2 outlines the rationale and approach for the Cartesian plot analysis here developed, and Section 4.7 in Materials and Methods provides details of methodology. In brief, by comparing the red and green fluorescence of co-cultured cells with that of control cells cultured alone, it was possible to unambiguously identify co-cultured cells as belonging to one of five separate groups of cells. In addition, the angular divergence of co-cultured cells from the origin and relative to the line bisecting the Cartesian plan, provided a measure of similarity in fluorescence of co-cultured cells to the two control cell populations that could be meaningfully compared amongst cells in what we defined as Exchange Units (EU). 

The five separate co-culture sub-populations as classified by Cartesian plot analysis (Figure 2) were seen in experiments. The results of an illustrative experiment are shown in Figure 3 with cells plotted on both linear (Figure 3a) and log (Figure 3b) scales to aid interpretation. The wide range of fluorescence in control cell populations cultured alone is evident in both the linear and log plots.

While the experiment shown in Figure 3 had cells in all five cell populations, this was not the case across all 24 co-cultures studied in experiments ‘a’ to ‘j’, and this is detailed in Table 1 as well as in MATLAB graphical outputs for all experiments shown in Appendix A. 

### 2.3. The Distribution of Co-Cultured Cells Classified According to Cartesian Plot Analysis Varied Amongst Co-Cultures, and Most Co-Cultured Cancer Cells Had Uptake of Fibroblast Fluorescence

In most co-cultures, the great majority of cancer cells accepted fibroblast fluorescence Table 1 (Figure 1, Figure 3, Figure 4 and Figure 5, Appendix A). Nonetheless, some cells with fluorescence indistinguishable from that in control cancer cells were seen in all experiments other than in experiment ‘c’, where all U2OS, MM200-B12, NM39, WM175, COLO316 and PEO1 were found to have received fluorescence from fibroblasts (Table 1, Figure 3, Figure 4 and Figure 5, Appendix A). 

While absolute certainty could not be had regarding the precise identity of individual co-cultured cells in the Cartesian plot classification group of ‘cells of uncertain origin’, considering the distribution of cells amongst classification groups, it seemed clear that the majority of these were cancer cells in most co-cultures, and this seemed especially likely where both groups were prominent that were defined as ‘cancer cells with some fibroblast labelling’ and ‘cells indistinguishable from fibroblast controls’ (Appendix A for SAOS-2 experiments ‘a’, ‘b’, ‘f’; Appendix A for U2OS experiment ‘a’; Appendix A for MeIRMu experiment ‘a’; Appendix A for MM200-B12 experiments ‘a’, ‘b’; ‘j’; Appendix A for WM175 experiment ‘a’; and Appendix A for Colo316 experiments ‘a’ and ‘b’). 

### 2.4. The Distribution of Co-Cultured Cells according to Exchange Units Varied amongst Co-Cultures

Variability was seen amongst co-cultures in the distribution of cells according to EU (Table 2). In 10 co-cultures, there was predominance of cells with fluorescence properties more akin to those of control fibroblasts, while the opposite was seen in six co-cultures where there was a preponderance of cells with fluorescence approaching or the same as that in control cancer cells (Table 2). ‘Fibroblast-like’ and ‘cancer cell-like’ cells each comprised close to 50% of cells in seven co-cultures (Table 2). In most of these co-cultures, comparatively few cells were in the mid-range group of −10 < EU ≤ 10, but in one case cells in the ‘mid range’ of EU approached in number those at the ‘cancer cell’ and ‘fibroblast’ poles (Table 2). 

All co-cultures had notable paucity of cells across several of the seven EU domains defined, so that with exception of one co-culture with SAOS-2, up to four of the EU groups defined above in each co-culture, had fewer than 5% of co-cultured cells, comprising 61 of all 168 EU domains across 24 co-cultures. 

### 2.5. FACS Demonstrated Preferential Transfer of Label from Fibroblasts to Cancer Cells Together with Higher Transferability of DiD Label Compared with DiO Label

It was noticed that in experiment ‘c’ where fibroblasts were labelled with DiO, uptake of fluorescence by fibroblasts from cancer cells was more prominent than in other experiments where fibroblasts were pre-labelled with DiO (Table 1 and Table 2; Appendix A). To explore the effect of swapping dye orientation on transfer, experiments were performed in which cancer cells (Colo316, SAOS-2, U2OS, MM200-B12, NM39 and WM175) were pre-labelled with either DiO or DiD, and then co-cultured with human dermal fibroblasts pre-labelled with opposing DiD or DiO before co-culture and fluorescence-activated cell sorting analysis (FACS). The resultant FACS plots are shown in Appendix A. 

In all co-cultures, there was transfer of fluorescence from fibroblasts to cancer cells to produce a ‘shifted’ cancer cell population in FACS, regardless if fibroblasts were labelled with DIO or DiD (Appendix A). However, uptake of cancer cell fluorescence by fibroblasts sufficient to produce a clearly ‘shifted’ fibroblast population, only occurred when cancer cells were pre-labelled with DiD (Appendix A). For reasons outlined in the discussion, it was concluded that orientating fluorescent labelling, such that DiD was in fibroblasts, would minimize ambiguity of results and facilitate interpretation. 

### 2.6. Cartesian Plot Analysis Indicated That Transfer of Fluorescence from Fibroblasts to Cancer Cells Was Associated with Reduced Cancer Cell Circularity and Increased Cancer Cell Profile Area

Uptake by cancer cells of fluorescence from fibroblasts was associated with reduced cell circularity and increased cell-profile area. Figure 3, panels ‘c’ and ‘d’ show an illustrative experiment where a dose-like response was seen between EU and both SAOS-2 circularity and cell-profile area. Comparable results for all remaining co-cultures in this study are seen in Appendix A. Figure 4 and Figure 5 summarise results for all 24 co-cultures across 10 experiments ‘a’ to ‘j’, relating cell circularity and cell-profile area to cell classification according to Cartesian plot (Figure 4), and EU (Figure 5). The results of statistical comparisons between groups of cells across all experiments shown in Figure 4 and Figure 5, are provided in Appendix A. Appendix A, also in Appendix A, show the results of the statistical evaluation of individual experiments.

Control cancer cells had higher cell-circularity and lower cell-profile area compared with control fibroblasts, and this difference was maintained when the cells were co-cultured (Figure 4, *p* < 0.0001, Wilcoxon signed-rank rest). Given the ambiguity regarding the precise identity of co-cultured cells in the Cartesian plot group ‘Uncertain Origin’, it was interesting to focus on the circularity and cell-profile area of those cells which were unambiguously cancer cells or fibroblasts, but with some fluorescence acquired from the opposing cell type (Figure 2, Figure 3 and Figure 4). For the same reason, it was similarly interesting to consider those cells in EU groupings immediately adjacent to EU = 50 and EU = −50 (Figure 5). 

In Figure 4a, co-cultured cancer cells with some fibroblast fluorescence had lower cell circularity compared with control cancer cells (*p* < 0.0001, Wilcoxon signed-rank test). However, while a similar difference in cell circularity seemed apparent comparing cancer cells with some fibroblast fluorescence against co-cultured cells with fluorescence indistinguishable from that in control cancer cells, this was not statistically significant. Separately, co-cultured cells with fluorescence indistinguishable from control fibroblasts were less circular, compared with co-cultured fibroblasts with some cancer cell fluorescence (Figure 4a, Appendix A, *p* < 0.03, Wilcoxon signed-rank test). Similar observations were made when cell-circularity of co-cultured cells was assessed against EU (Figure 5a, Appendix A). 

Results for cell-profile area across all co-cultures are given in Figure 4b and Figure 5b (Appendix A), and show a similar effect for cell-profile area as was seen for circularity, where uptake of fibroblast fluorescence by cancer cells was associated with increased cell-profile area (*p* < 0.042, Wilcoxon signed-rank test). SAOS-2 were studied in 9 of the 10 experiments, so it is interesting to consider these cells alone. Results for SAOS-2 were broadly similar to those considering all cancer cell lines together, and details of statistical comparisons are given in Appendix A. 

While there was variability amongst co-cultures, and it was common for one or more of the groups of cells classified according to Cartesian plot or by EU to be absent, the observations outlined above were seen across most individual co-cultures. Details for individual co-cultures of observations on fundamental differences in cell circularity and cell-profile area between fibroblasts and cancer cells are given in Appendix A, while numerical analysis of this is in Appendix A. Similarly, details for individual co-cultures of observations relating to cell circularity with reference to Cartesian plot classification and EU are in Appendix A, respectively, while Appendix A provides a numerical summary; similarly, Appendix A provide a parallel description of data for cell-profile area. 

Taken together, these observations, indicate a dose–response-like effect of uptake of fibroblast fluorescent contents, on cancer cell morphology. 

### 2.7. The Orientation of Fluorescent Dyes in Co-Cultures Did Not Affect Results of Cartesian Plot Analysis with Regard to Morphological Phenotype

As seen in Cartesian plot analysis of experiment ‘c’ (Table 1 and Table 2; Figure 4 and Figure 5; Appendix A), as well as by FACS analysis (Appendix A), when cancer cells were pre-labelled with DiD there was much higher transfer of fluorescent labels from co-cultured cancer cells to fibroblasts than occurred when cancer cells were labelled with either DiO or GFP. This dye orientation had no clear effect on Cartesian plot analysis results for morphological phenotype, such that cancer cell circularity reduced, and cancer cell surface profile area increased on uptake of DiO fluorescence from fibroblasts (Appendix A). Similarly, there was no clear effect on results where SAOS-2 were labelled with GFP in experiment ‘i’, instead of with DiO in other experiments (Figure 4 and Figure 5; Appendix A). 

### 2.8. Nuclei Labelled for 5mc and Propidium Iodide

In experiments ‘m’, ‘n’, ‘o’ and ‘p’, the appearance of co-cultured cells as well as of the distribution of DiD and DiO labels were as seen in the preceding experiments described above. Propidium iodide clearly identified nuclei, as expected from the intended DNA labelling. HDF in control cultures appeared to have less intense fluorescence for propidium iodide, compared with many control SAOS-2 (Appendix A). Labelling for 5mc was characteristically nuclear, and generally less uniform within nuclei compared with propidium iodide labelling, consistent with localised collections of DNA with high 5mc. Comparing fibroblasts in co-cultures with control fibroblasts cultured in isolation, the visual impression was that co-culture reduced 5mc fluorescence (Figure 6 and Appendix A). 

### 2.9. Cell Circularity and Cell-Profile Area Were as Expected

Appendix A shows typical results of Cartesian plot analysis, in this instance from experiment ‘n’. As demonstrated above, fibroblasts had a higher cell-profile area and lower cell circularity compared with SAOS-2 (Appendix A). Considering co-cultures, associations between cell-profile area and cell circularity, and identity of cells as belonging to one of the five Cartesian plot groupings was also as described above, and the same applied when considering EU (Appendix A). 

### 2.10. Control HDF Cultured in Isolation Had Higher Normalised 5mc Compared with Control SAOS-2

Figure 7 shows Cartesian plot distributions for normalised 5mc according to EU, while the Cartesian plot grouping of individual cells is represented by colour. As expected from the numerical method used to normalise corrected nuclear fluorescence (cnf) (see 4.11 and 4.12 in Materials and Methods), the median values for control fibroblasts were at or very close to 100, with slight divergence accounted for by the handling of outliers for control cnf specified in the relevant MATLAB script. Control SAOS-2, had low 5mc relative to HDF across all experiments (*p* < 0.035, paired *t*-test considering all experiments together; *p* < 0.015, Mann–Whitney U tests within individual experiments). Numerical results and statistical details are summarised in Appendix A.

### 2.11. Co-cultured Cells with DiD–DiO Fluorescence Indistinguishable from Control HDF, Had Fluorescence Lower than HDF Control Cells

Co-cultured cells with DiD–DiO fluorescence indistinguishable from control HDF were considered to be unambiguous fibroblasts in co-culture. Figure 7 shows that across all co-cultures studied, these cells had appreciably lower normalised 5mc compared with the control HDF cultured in isolation (*p* < 0.25, paired *t*-test considering all experiments together; *p* < 0.0025, Mann–Whitney U tests within individual experiments). Numerical results and statistical details are summarised in Appendix A.

### 2.12. There was No Convincing Relationship between Cell-Projection Pumping and Changes in Normalised 5mc of Co-Cultured Cells

While the effects of co-culture were seen on overall 5mc levels relative to controls, there was no clear ‘dose-like’ effect on normalised 5mc related to either the Cartesian plot grouping or EU (Figure 7; Appendix A). 

A tendency for co-cultured cells to have still lower normalised 5mc relative to control SAOS-2 cultured in isolation did seem apparent in experiments ‘m’ and ‘n’, but this seemed countered by a strongly opposing pattern in experiment ‘o’, and indifferent distinction in experiment ‘p’ (Figure 7). Similarly, a slight ‘dose-like’ effect did seem apparent for median EU values in experiment ‘o’ (Figure 7); however, the very small number of cells in each EU group deeply undermines confidence in this possible relationship, and this doubt is further reinforced by the absence of a similar effect in remaining experiments ‘m’, ‘n’ and ‘p’. 

## 3. Discussion

There was great variation amongst co-cultures with regard to the extent of exchange between cells and the distribution of cells according to Cartesian plot group classification and/or EU. The spread of co-cultured cells across Cartesian plots undermined numeric analysis, as did the extent to which initial fluorescence amongst control cells varied. Nonetheless, despite this high variability, it was possible to make statistically meaningful statements on the relationship between morphological phenotype and fluorescence transfer. Confidence in this approach is given by consistency of the current findings with those of separate earlier work, where increased SAOS-2 cell profile area and reduced cell circularity was seen with uptake of fluorescence from fibroblasts; determined in fixed monolayers at the cell population level [1].

While earlier morphological analysis was of SAOS-2 only [1], the current work shows similar effects in other cell lines, including in the separate U2OS osteosarcoma cell line, as well as melanoma, colon cancer and ovarian cancer cells. This suggests that the effects of uptake of fibroblast contents on cancer cell morphology are not idiosyncratic to SAOS-2, but are typical of a generalised response of cancer cells, strengthening the sense that intercellular transfer of cellular material by cell-projection pumping plays a role in generating cancer cell diversity [1,2,3]. Certainly, the current work demonstrates increased morphological diversity amongst cancer cells upon uptake of fluorescence from fibroblasts, and it is separately accepted that cancer cell diversity drives clinical disease progression and is an important morphological prognostic marker in histopathology [31]. 

It was interesting that altered fibroblast morphology in response to uptake of cancer cell fluorescence was also seen, and this may have bearing on the emergence of cancer associated fibroblasts, increasingly thought important in cancer biology [27,51].

It was surprising to find that the orientation of fluorescent dye had an effect on patterns of fluorescence distribution between co-cultured cells, such that DiD more readily passed between cells. This first became apparent through Cartesian plot analysis, and was then confirmed by FACS. DiD has high lateral plasma membrane mobility compared with DiO [52], so it is concluded that DiD readily traverses intercellular continuities via the contiguous plasma membrane from any DiD pre-labelled cell during cell-projection pumping or via tunnelling nanotubes, while transfer of fluorescently labelled cytoplasmic organelles is predominantly from human dermal fibroblasts to cancer cells by cell-projection pumping. Current Cartesian plot and FACS analysis [2] improves quantitation over the earlier microscopy study with fixed monolayers [1], but is consistent with the previously reported microscopic observations of: DiD and DiO exchange; transfer of DDAO-SE labelled proteins from HGF into GFP expressing SAOS-2; transfer of CFSE from HGF into DDAO-SE labelled MC; and the transfer of plasma membrane alkaline phosphatase with high lateral membrane mobility from SAOS-2 to HGF [1,53]. Despite asymmetry in transferability of DiD and DiO, the relationship between fluorescence transfer and cancer cell morphology was unaffected, while the use of GFP as the cancer cell marker also had no effect on the morphological changes seen. 

The current work establishes what appears to be a novel approach to quantitation of fluorescence transfer between populations of co-cultured cells. This offers the opportunity to relate the expression of separate phenotypic markers to the extent of cytoplasmic transfer, including, for example, levels or patterns of cytoskeletal elements, organelles, or any other cell component that can be fluorescently labelled. The method appears reasonably robust, as it was insensitive to differences in the application of fluorescent markers, and generated results consistent with earlier work [1].

Of the two means of evaluating fluorescence transfer by Cartesian plot analysis trialed, the preference is for use of Cartesian plot classification groups, because this defines unambiguous sub-populations of cells that can be compared with each other with relative confidence. However, there is value in also using EU, especially where there are few if any cells in the two groups comprising ‘red’ or ‘green’ cells, ‘with some fluorescence’ from the opposing cell type. 

As outlined above, cancer cell morphology is considered important for cancer diagnosis and assessing clinical grade [54,55]. Assuming that morphological diversity amongst cultured cancer cells reflects morphological diversity amongst cancer cells in vivo, the current findings support the idea that cell-projection pumping contributes to clinically significant cancer cell diversity in vivo. 

To detect 5mc in situ as in the current work, it was necessary to permeabilise cells with powerful detergents, which necessarily degrade cell membrane structures. Because DiD and DiO are both lipophilic markers that bind to cell membranes, it was necessary to record DiD and DiO fluorescence prior to cell permeabilisation for 5mc detection. This added to the complexity of the methodology and analysis, because in addition to recording across four separate wavelengths, it was necessary to record different labels on two separate occasions, and address the difficulty of accurately overlaying images. In addition, variability inherent to in situ labelling systems, made the method developed highly technique sensitive. Despite these challenges, data were generated that could be analysed in a meaningful way, and could also be subjected to meaningful statistical analysis. 

The expression of 5mc fluorescence relative to total DNA fluorescence and then normalisation relative to control HDF provided a convenient and justifiable means for quantitation of 5mc that reduced sensitivity to the inherently variable nature of immuno-histochemistry, and also permitted direct comparison of results from experiments conducted on different days. Confidence is built in this approach by the high statistical significance of many of the results obtained within experiments, as well as by the ability to make statistically meaningful statements when all experiments were considered together. 

One clear observation to emerge from the experiments shown was that fibroblasts have higher normalised 5mc compared with SAOS-2. This is consistent with the work of others, who report hypomethylation of DNA in cancer cells relative to normal cells [36,56,57,58,59,60]. A further clear observation, was that fibroblasts co-cultured with SAOS-2 have markedly lower normalised 5mc compared with control fibroblasts cultured alone, and that this approaches levels seen in SAOS-2. We have not characterised the fibroblasts in our co-cultures to determine if their phenotype is in alignment with that of cancer associated fibroblasts. However, the lower normalised 5mc of co-cultured fibroblasts is consistent with the separate literature where cancer associated fibroblasts are reported as hypomethylated compared with normal fibroblasts [47,48]. This finding in the current work suggests that the co-culture system used may provide a good model for generating artificial cancer associated fibroblasts for separate in vitro study, but further work beyond the scope of the current investigation would be required to determine if this is the case. Irrespective, consistency of lower normalised 5mc in co-cultured fibroblasts with low 5mc found in cancer associated fibroblasts, does lend further confidence in the utility of the Cartesian plot method. 

It has been reasonable to speculate that cell-projection pumping might play a role in the development of cancer associated fibroblasts, especially since some transfer from SAOS-2 to fibroblasts has been reported in earlier work from this laboratory [1,2]. However, despite seeing morphological changes in co-cultured SAOS-2 consistent with cell-projection pumping, there was no similar correlation between normalised 5mc in either SAOS-2 or HDF, and transfer of fluorescence between the two cell populations. From this, the possibility that cell-projection pumping drives cancer associated fibroblast differentiation is in large part refuted by the current results. In light of the current data, it seems more likely that cancer associated fibroblast differentiation is driven by soluble factors and/or exosomes released by cancer cells, and this is consistent with the literature [61]. Nonetheless, the possibility that cell-projection pumping might contribute to cancer associated fibroblast differentiation, could be further studied using the Cartesian plot method to correlate the expression of cancer associated fibroblast markers, such as smooth muscle cell actin, with fluorescence uptake from cancer cells.

Similarly, these data provide no evidence for a role of cell-projection pumping from fibroblasts in determining global DNA methylation status of cancer cells. On balance, data strongly suggest the opposite conclusion. 

By essentially excluding cell-projection pumping from having a role in determining global DNA methylation in co-cultures, the current study demonstrates general utility applying the Cartesian plot method to phenotypic characteristics separate to the morphological properties of cell-profile area and cell circularity. Cell-profile area always provides a convenient ‘internal control’ for phenotypic effect of cell-projection pumping studied by Cartesian plot analysis, while as here demonstrated, numerical analysis for any other phenotypic trait that can be localised to individual cells, is readily achieved by simple substitution for cell circularity in the MATLAB script provided. 

Despite the limitations inherent to the study of fixed cell monolayers, the Cartesian plot method has the advantage of relative speed compared with the other single-cell analytical method we have used, single-cell tracking [4], and further permits study of the relationship between cell-projection pumping and possible phenotypic effects, which are impossible to examine in viable cells. 

We have considered if inhibition of cell-projection pumping might have therapeutic value treating cancer, by reducing cancer cell diversity [1,3,4,30]. Recent single-cell tracking studies suggest that the phenotypic effects on cancer cells of uptake of fibroblast contents by cell-projection pumping are lost upon cancer cell division, effectively re-setting daughter cells to their native cancer cell phenotype [4]. The current observation that there is no clear effect of cell-projection pumping on cancer cell global DNA methylation, supports the idea that cell-projection pumping does not cause epigenetic change in cancer cells. This has therapeutic significance, because any agents inhibiting cell-projection pumping would have only limited value reducing cancer cell diversity, if the cancer cell diversity generated by cell-projection pumping were ‘locked in’ by epigenetic effects. 

## 4. Materials and Methods

### 4.1. Experiments across Two Laboratories

Sixteen experiments coded ‘a’ to ‘p’ were performed in two separate laboratories, being the laboratory of Dr MAS Moore at the Memorial Sloan Kettering Cancer Center (MSKCC) New York (experiments a, b, i, j, k, l), and The Cellular and Molecular Pathology Research Unit (CMPRU) in the University of Sydney (experiments c, d, e, f, g, h, m, n, o, *p*). Because of this, the source of materials varied according to experimental location. 

### 4.2. Materials

#### 4.2.1. Materials for Cartesian Plot Experiments Performed at MSKCC

For experiments performed at MSKCC, all culture media including M199, α-MEM, Trypsin (0.25%)-EDTA (0.02%) and PBS, as well as Penicillin (10,000 U/mL)-Streptomycin (10,000 μg/mL) concentrate solution were prepared and supplied by the Memorial Sloan-Kettering Cancer Centre Culture Media Core Facility (New York, NY, USA). Amphotericin B was purchased from Life Technologies (Grand Island, NY, USA). Gelatin was from TJ Baker Inc. (Philipsburgh, NJ, USA). Bovine serum albumin was from Gemini Bioproducts (West Sacramento, CA, USA). Falcon tissue culture flasks and centrifuge tubes were purchased from BD Biosciences (Two Oak Park, Bedford, MA). Nunc plastic culture well coverslips were from ThermoFisher, Waltham, MA, USA). The lipophilic fluorescent probes DiD (excitation 644 nm, emission 665 nm) and DiO (excitation 484 nm, emission 501 nm) Vybrant cell labelling solutions were from Molecular Probes, Life Technologies (Grand Island, NY, USA), and were used in all experiments at both the MSKCC and CMPRU. 

#### 4.2.2. Materials for Cartesian Plot Experiments Performed at the CMPRU

For experiments performed in the CMPRU, all cell culture media, were obtained from Sigma-Aldrich (St. Louis, MO, USA). Bovogen (Keilor East, VIC, Australia) supplied iron fortified bovine calf serum (BCS). JRH Biosciences (Lenexa, KS, USA) supplied Trypsin (0.25%)/EDTA (1 mM). CSL Biosciences (Parkville, VIC, Australia) supplied antibiotics penicillin and streptomycin. ICN Biomedicals Inc. (Solon, OH, USA) supplied bovine serum albumin (BSA) fraction V and amphotericin B. Phosphate buffered saline (PBS) tablets were from Oxoid (Hampshire, UK). Iwaki, Scitech Division (Chiba, Japan) supplied centrifuge tubes. Costar (Cambridge, MA, USA) supplied tissue culture plastic-ware. Glass tissue culture slides were used in experiments for Cartesian plot analysis (ThermoFisher, Waltham, MA, USA). 

For experiments investigating global DNA methylation, PLL coated high performance 12 mm coverslips available from Neuvitro Corporation (Camas, Waltham, WA, USA), were used for cell culture, while Superfrost^®^ Plus adhesive microscope slides (Fisher Scientific, MA, USA) were used to support tissue culture coverslips for microscopy. Mouse monoclonal anti 5-Methylcytidine antibody was obtained from Biorad Laboratories (Hercules, CA, USA). Goat anti-Mouse IgG (H + L) Cross-Adsorbed Secondary Antibody labelled with Alexa Fluor 405 (excitation 401 nm, maximum emission 421 nm) was purchased from Thermo Fisher Scientific (Waltham, MA, USA), as was goat serum, 0.25 % Trypsin/EDTA, propidium iodide (maximum excitation 535 nm and maximum emission 617 nm), and SlowFade Diamond Antifade: soft-setting mountant. Bovine serum albumin, hydrochloric acid 37% and sodium azide were from Sigma Aldrich (St Louis, MO, USA). Cold water fish skin gelatin was from Merck (Kenilworth, NJ, USA). Acetylated BSA (10% BSA-c) was purchased from AURION (Wageningen, The Netherlands). 

#### 4.2.3. Cells Used in Experiments

Human dermal fibroblasts from The Coriell Institute (Camden, NJ, USA) were used in experiments performed at MSKCC, and from the American Type Culture Collection (Manassas, VA, USA) in experiments conducted at the CMPRU. The only exception was for experiment ‘c’ in the CMPRU, where human gingival fibroblasts were used and obtained from explant cultures as previously described [1]. SAOS-2 osteosarcoma cells were from the American Type Culture Collection (Manassas, VA, USA), and a SAOS-2 cell line expressing GFP was as earlier reported and used in experiment ‘i’ [1]. Further cancer cell lines studied were from the collection at the Westmead Institute of Medical Research (Westmead, NSW, Australia) and shared between collaborating laboratories of the CMPRU and MSKCC. 

### 4.3. Cell Culture

The antibiotics penicillin (100 U/mL), streptomycin (100 µg/mL) and amphotericin B (2.5 µg/mL) were used throughout all cell culture. Culture conditions differed according to cell type, such that: fibroblasts were always cultured on gelatin coated surfaces (0.1% in PBS) in alpha-MEM (15% FCS at MSKCC or BCS in the CMPRU); SAOS-2 and U2OS were cultured in M199 with Earl’s salts (10% FCS at MSKCC or BCS in the CMPRU); Colo 316 and PEO1 were cultured in RPMI1640 with 15 mM HEPES (10% FCS at MSKCC or BCS in the CMPRU); MeIRMu, MM200-B12, NM176, NM39 and WM175 were cultured in DMEM (10% FCS at MSKCC or BCS in the CMPRU). Cells were harvested using trypsin-EDTA, into FCS at MSKCC or BCS at the CMPRU to neutralise trypsin, and pelleted by centrifugation before passage at a ratio of 1 to 3. All cell culture was performed at 37 °C under CO_2_ (5%) and at 100% humidity. 

### 4.4. Labelling of Cells with Lipophilic Fluorescent Membrane Markers

Labelling solutions of DiD (1 mM) and DiO (2 mM) were prepared in alpha-MEM with 10% FCS, and applied to cells for 30 min in the case of DiD, while DiO was applied for 1 h. Cells were then washed twice with PBS before overnight culture with alpha-MEM with BSA (4%) followed by two further washes with PBS in order to ensure the removal of any unbound label [1,2,3]. Most experiments were conducted pre-labelling fibroblasts with DiD (experiments ‘a, b, d, e, f, g, h and i’). Similarly, DiO was usually used to pre-label cancer cell lines (experiments a, b, d, e, f, g, h, j, k, l, m, n). In one experiment cancer cells were labelled with DiD and human gingival fibroblasts were labelled with DiO (experiment ‘c’). In one further experiment, SAOS-2 expressing GFP were used (experiment ‘i’); thus not requiring DiO labelling. In the two experiments for FACS analysis (experiments ‘k’ and ‘l’), the effect of swapping dye orientation between human dermal fibroblasts and cancer cells was examined. 

### 4.5. Co-Culture Conditions 

A total of 34 co-cultures were conducted across 16 experiments. All experiments were with cells cultured on gelatin coated surfaces (0.1% in PBS). Fibroblasts were seeded at from 1 to 2 × 10^4^ cells per cm^2^ and allowed to adhere overnight before labelling with either DiD or DiO and further overnight culture in alpha-MEM with BSA (4%) as outlined above. With the exception of a single experiment ‘c’ where human gingival fibroblasts were used, all experiments were conducted using human dermal fibroblasts. Cancer cells were seeded prior to labelling at near confluence in culture media appropriate to the cancer cell line, and allowed to adhere overnight before labelling with DiD or DiO and further overnight culture in alpha-MEM with BSA (4%) as outlined above. Pre-labelled cancer cells were then seeded in alpha-MEM with BSA (4%) at a culture density of 4 × 10^4^ cells per cm^2^ for 24 h co-culture. Control cultures comprised fibroblasts and cancer cells seeded and labelled in parallel. Cells for Cartesian plot analysis were cultured on tissue culture slides and fixed with paraformaldehyde (4% in PBS) before washing in PBS, while those for FACS analysis were cultured in 12 well tissue culture plates. 

### 4.6. Quantitation of Fluorescence in Fixed Adherent Cells and Measurement of Cell Circularity and Cell Profile Area

In the course of this study, three separate methods were applied to segment cells, and these produced comparable results; although the ease of application increased with development of the methodology. In the earliest experiment (experiment ‘c’), segmentation of cells in two-channel fluorescence images was entirely manual, using ImageJ Version 1.48, accessed June 2014 and developed at NIH with Wayne Rasband as project developer at the Laboratory for Optical and Computational Instrumentation, University of Wisconsin, Madison, WI, USA as open source software available at http://imagej.nih.gov/ij. This limited the number of cells that were practical to sample, and so a separate semi-automated approach was developed using script also in ImageJ, in which individual cells were segmented on the basis of colour threshold in the green and then red channels, before being combined into single objects. This method was applied in experiments a, b, i, and j, and required manual separation of closely adjacent cells. A more effective approach was later developed in MATLAB script (MathWorks, Natick, MA, USA) and this was used in all remaining experiments, with the advantage of improved automated separation of cells with less manual correction; thus permitting more cells to be sampled. Red and green fluorescence was quantitated in segmented cell areas. Cell-profile area and cell perimeter were also determined for segmented cells, and these were used to calculate cell circularity, which is a unitless measure calculated by the equation: cell circularity = 4 pi(Cell profile area)/(cell peripheral circumference)^2^.

### 4.7. The Cartesian Plot Method for Quantitation of Fluorescence Transfer

Figure 2 illustrates the rationale and approach for the Cartesian plot analysis. In brief, image analysis gave green and red fluorescence, as well as cell-profile area and circularity for individual control cell cultured in isolation, as well as for co-cultured cells. Up to 1000 co-cultured cells were included in the analysis, while 100 control cells were found to be convenient for defining fluorescence boundaries without undue influence of outliers. Background fluorescence was determined by quantitation of red and green fluorescence per unit area in summated red and green labelled control cells. Background was subtracted in relevant channels on the basis of cell-profile area, before normalisation of fluorescence for all cells by comparison with control cell populations cultured alone, defining a value of ‘100 normalised fluorescence units’ for the median fluorescence of control cells. Each cell was then plotted against normalised fluorescence of control cells cultured alone, with ‘red control’ cells aligned on the *y* axis, and ‘green control cells’ placed on the *x* axis. The location in Cartesian plots identified co-cultured cells as belonging to one of five cell classifications illustrated in Figure 2, which could be compared against each other. 

A further assessment of cells was made in which individual co-cultured cells were assigned ‘Exchange Units’ (EU) representative of the degree of fluorescence similarity with control cells cultured in isolation. It was argued that ‘Complete mixing’ of fluorescence between cell populations would cluster cells about the diagonal dashed line shown in Figure 2. This permitted the determination of a measure for transfer to individual cells, by first subtending a line to the origin from any plotted cell, and then measuring the angle of divergence relative to the dashed diagonal line (Figure 2). *x* and *y* axes were assigned 50 and −50 Exchange Units, respectively, such that cells along the diagonal had ‘0 Exchange Units’, and cells below or above the diagonal approached 50 or −50 EU according to the angular relation to the *x* and *y* axes, respectively. It was convenient to consider co-cultured cells in the following 7 groups according to EU: EU = −50; −50 < EU ≤ −30; −30 < EU ≤ −10; −10 < EU ≤ 10; 10 < EU ≤ 30; 30 < EU ≤ 50; EU = 50. 

### 4.8. Automation of the Cartesian Plot Analysis Method by MATLAB Script and Statistical Evaluation

The Cartesian plot analysis was automated with script developed in MATLAB, which took as inputs data on: segmented cell DiO and DiD fluorescence; cell-profile area; and cell circularity. The relevant MATLAB script is provided in Appendix A, and can be readily adapted for use in other systems. Using both of these approaches for Cartesian plot analysis, it was possible to relate the transfer of fluorescence between cell populations to morphological features at the single cell level (Figure 1). Statistical analysis was with Prism software (9.2.0, Graphpad Software, San Diego, CA, USA), using the Mann–Whitney U test to compare groups within experiments, and the Wilcoxon signed-rank test to evaluate paired results across experiments ‘a’ to ‘j’. While we preferred these non-parametric tests on the basis that no assumption on distribution of data was required, the small sample size available in the four experiments ‘m’ to ‘p’ where 5mc levels were evaluated necessitated our application of the less conservative paired *t*-test. 

### 4.9. Fluorescence Activated Cell Sorting Analysis

Human dermal fibroblasts and the cancer cell lines (Colo316, SAOS-2, MM200-B12 in experiment ‘k’; U2OS, NM39, and WMM175 in experiment ‘l’), were either pre-labelled with DiO or DiD, or left unlabelled as fluorescence controls in 25 cm^2^ culture flasks on gelatin. Please note that labelling was arranged such that for each co-culture, there was direct comparison of opposite labelling orientation, with fibroblasts and cancer cells labelled each with DiD or DiO, and 24 h isolated cell control cultures and co-cultures were established for all cells, before washing and harvesting with trypsin-EDTA. Cells were fixed by dropwise addition of paraformaldehyde (4% in PBS), washing by centrifugation and resuspension in PBS, and protected from light before being subjected to FACS analysis using a LSF Fortessa analyser and Flowjo software (BD Biosciences, Fanklin Lakes, NJ, USA).

### 4.10. Labelling to Relate Global DNA Methylation with DiO/DiD Fluorescence in Co-Cultures

Fixed monolayers were protected from light at all times. Fluorescence images for DiO and DiO were recorded as described above, prior to labelling for 5mc. In brief, tissue culture coverslips were removed from microscope slides by immersion in PBS for 15 min, and then gentle prying of coverslips off microscope slides using an 18G syringe needle. Coverslips were then washed three times with PBS. 

Because 5mc quantitation was necessary for meaningful analysis, correction was needed for total DNA in cells, allowing for variation in the amount of DNA through the cell cycle as well as for aneuploidy in SAOS-2 [62]. For this reason, in addition to detecting fluorescence for antibody labelling of 5mc using a method adapted from that described by others [63], cells were also labelled for nuclear DNA using propidium iodide as earlier described [62]. 

Prior to 5mc labelling, a ‘blocking buffer’ was prepared by combining 2.5 g BSA, 2.5 mL of normal goat serum, 100 µL cold water fish skin gelatin (1% in PBS), 2.5 mL of sodium azide (200 mM) in 30 mL PBS. The pH was adjusted to 7.4. An incubation buffer was prepared by combining 1 mL acetylated BSA (10%), 2.5 mL sodium azide (200 mM) in 35 mL of PBS. The pH was adjusted to 7.4 for the final solution. Tissue cover coverslips were transferred to wells in 6 well tissue culture plates, and monolayers were incubated with glycine (2% in PBS) for 10 min and then washed twice with DMEM for 5 min each. Coverslips were then incubated in Triton X100 (0.75% in PBS) together with 0.75%Tween 20 (0.75% in PBS) for 5 min to permeabilise the cell for 5 min, before washing in DMEM twice for 5 min each. Chromatin was denatured by incubation in 4N HCL for 10 min, followed by 1 min treatment with 0.25% Trypsin/EDTA. Enzyme activity was blocked by the application of goat serum (10% in DMEM) followed by washing 3 times for 5 min each with DMEM. 

Cells were incubated for 1 h at RT with blocking buffer to minimise non-specific binding. Coverslips were then washed 3 times with Tween-20 (0.02% in PBS) followed by the application of mouse anti-5mc antibody (1:100 in PBS with 0.2% Tween-20) and overnight incubation at 4 °C in a humidification chamber. Tissue coverslips were then washed three times for 5 min each in Tween-20 (0.03% in PBS) before the application of fluorescent Goat anti-Mouse antibody (1:200 dilution in incubation buffer) for 2 h at RT in a humidified chamber. Monolayers were then washed 3 times with Tween-20 (0.03% in PBS) for 5 min each, before dipping in distilled water, and labelling for nucleic acid using propidium iodide (1 mM in PBS) for 5 min. Coverslips were then washed with DMEM three times for 5 min each before imaging. Controls comprised cells processed in parallel, but with the exclusion of the secondary antibody. 

Imaging was with a VS120 automated slide scanner (Olympus Lifesciences, Tokyo, Japan) using Cell Sense Software (Olympus Lifescience, Japan), using the DAPI filter to visualise anti-5mc label, and the ‘TX-RED filter’ for propidium iodide. 

Images in .vsi format were converted to TIFF format using Olyvia software (Olympus life science, Tokyo, Japan). Subsequent images of DiO and DiD fluorescence were separately overlayed with images for nuclear labelling with anti-5mc and propidium iodide using Adobe photoshop (San Jose, CA, USA). Cell segmentation and quantitation of DiO, DiD, cell circularity and cell-profile area were as described above. Nuclear segmentation and quantitation of 5mc and propidium iodide fluorescence was performed in a similar manner, with the exception that, respectively, different images were used. Data were collated to integrate all fluorescence and morphological properties of interest for each segmented cell. 

### 4.11. Normalisation of 5mc Fluorescence Relative to Propidium Iodide Fluorescence and Fibroblast Controls

As mentioned above, it was important to correct 5mc fluorescence for the total DNA content in individual nuclei. In order to facilitate the comparison of results across experiments, it was further helpful to normalise the corrected 5mc fluorescence values relative to a reasonable assumed constant. It was decided that HDF nuclear fluorescence would comprise the most stable and reproducible standard, against which all nuclear fluorescence in the remainder of each individual experiment could be measured. 

For the above reasons, nuclear 5mc fluorescence for each cell was first corrected by division by propidium iodide fluorescence of the same cell, giving a corrected nuclear fluorescence (cnf). The cnf value for each cell was then further normalised relative to HDF within the experiment to which the cell belonged, as per: Normalised cnf = (100 × cnf)/(median cnf of HDF Control Cells). 

### 4.12. Numerical Analysis of Global 5mc Fluorescence Related to the Cartesian Plot 

Cartesian plot analysis identical to that described above was performed for each experiment (m, n, o, p) using the MATLAB script given in Appendix A. This script and input files were slightly modified for analysis of normalised cnf, replacing data for cell circularity in input files, with that for normalised cnf, and also permitting the *y* axis of the graphical plot for display of cell circularity to extend beyond 1 to any normalized cnf value convenient for the given experiment. Statistical analysis was with Prism software (9.2.0; Graphpad Software, San Diego, CA, USA), using the paired t-test for comparisons across groups for all experiments, and the Mann–Whitney U test for comparisons across groups within experiments, with *p* < 0.05 considered statistically significant. 

## 5. Conclusions

The Cartesian plot analysis developed in this paper can be applied in the study of fixed monolayers of cells to identify sub-populations of co-cultured cells that have acquired fluorescence from the opposing cell culture partner, and also to relate fluorescence transfers to changes in phenotype. Transfer of fluorescent label between fibroblasts and cancer cells affects cell morphology. Fibroblasts have higher levels of DNA methylation compared with SAOS-2, and fibroblasts co-cultured with SAOS-2 have reduced global DNA methylation relative to control fibroblasts cultured alone, similar to cancer associated fibroblasts. This suggests that SAOS-2–fibroblast co-cultures, may comprise a convenient in vitro method for generating cancer associated fibroblasts. However, the absence of a relationship between fluorescence transfers or global DNA methylation levels in either fibroblasts or SAOS-2, suggest that cell-projection pumping does not contribute to either emergence of cancer associated fibroblasts, or altered DNA methylation in cancer cells. This supports the potential value of therapeutic agents inhibiting cell-projection pumping with the intention of reducing cancer cell diversity.

## Figures and Tables

**Figure 1 ijms-23-07949-f001:**
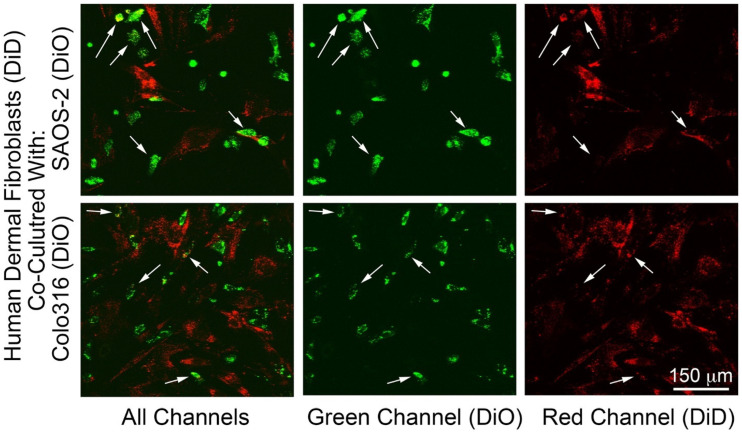
Fluorescence images of co-cultures of human dermal fibroblasts pre-labelled with DiD (red) with either SAOS-2 or Colo316 cell lines pre-labelled with DiO (Green), showing all channels combined, as well as the green (DiO) and red (DiD) channels alone. As expected for these lipophilic markers, fluorescence was concentrated in organelles to produce a punctate appearance. Fibroblasts were generally larger and less circular compared with the co-cultured cancer cells. Some SAOS-2 and Colo316, had evident uptake of fluorescent label from fibroblasts (white arrows).

**Figure 2 ijms-23-07949-f002:**
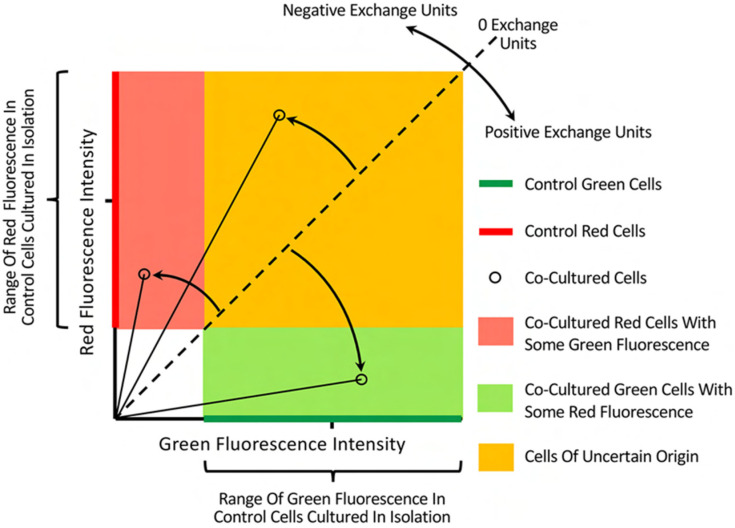
Diagram outlining Cartesian plot analysis of fluorescence transfer in co-cultures. Green and red fluorescence of cells in control populations cultured alone, was recorded along the x and y axes, respectively. The location of individual co-cultured cells was then defined by green and red fluorescence in the cartesian plane, and their position relative to control cells identified them as belonging to one of five separate populations in the co-culture. Co-cultured cells with fluorescence indistinguishable from that of control populations were defined as being either ‘red’ or ‘green’ cells, respectively. Co-cultured cells with fluorescence within the range of one of the control cell populations, but less than the fluorescence of the opposing cell type, were identified as belonging to either the red or green cell population, but having acquired some fluorescence from the opposing co-cultured cells. Co-cultured cells with fluorescence that overlapped with that of both control cell populations, were defined as ‘cells of uncertain origin’. The wide range of fluorescence values in control cells, undermined meaningful quantitation of fluorescence transfer from direct fluorescence measures. To overcome this, an angular measure for fluorescence transfer was derived. We reasoned that if all red and green cells achieved complete mixing, then all co-cultured cells would cluster about the dashed diagonal line at Green = Red fluorescence. Angular divergence from this diagonal of lines subtended from individual cells to the origin, gave a measure of similarity to either of the two control cell populations. Cells on the Diagonal were assigned ‘0 Exchange Units’, while control green and red cells were assigned 50 and −50 Exchange Units, respectively. Two cells above the diagonal are shown with negative Exchange Units, and are identified as belonging to two separate cell populations. One cell below the diagonal line has transfer measured in positive Exchange Units, and belongs to the population of green cells that have received some red fluorescent label.

**Figure 3 ijms-23-07949-f003:**
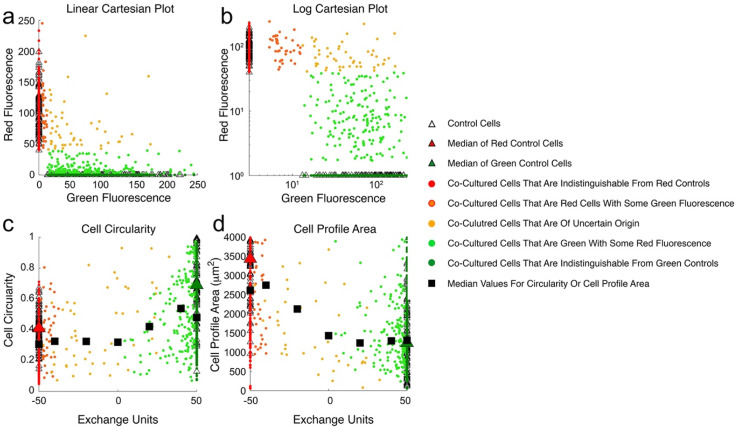
Cartesian plot analysis for cell circularity and cell surface area profile of an illustrative experiment (experiment ‘f’) co-culturing SAOS-2 pre-labelled with fluorescent green DiO, together with human dermal fibroblasts pre-labelled with fluorescent red DiD. Cartesian plots applying both linear (**a**) and log-scales (**b**) are shown, and the location of control cells is shown with open triangles. Cell circularity (**c**) and cell profile area (**d**) are shown for both control and co-cultured cells plotted against Exchange Units (EU). Medians are shown for both control cell populations (red and green triangles), as well as for co-cultured cells in groups of 20 ascending EU, but excluding those cells with fluorescence indistinguishable from controls. (**a**) Co-cultured cells belonging to all five cell populations as defined by the Cartesian plot method were seen in this experiment; (**b**) these are most easily distinguished when examining data in a log–log plot; (**c**) cell circularity of control SAOS-2 was greater than that of control fibroblasts (*p* < 0.0001, Mann–Whitney U test). Cell circularity was lower in all groups of co-cultured cells compared with control SAOS-2 (*p* < 0.025, Mann–Whitney U test); (**d**) cell-profile area of control SAOS-2 was less than that of fibroblasts (*p* < 0.0001, Mann–Whitney U test), and with the exception of cells with −50 < EU < −30, cell profile area was higher in co-cultured cells compared with control SAOS-2 (*p* < 0.0035, Mann–Whitney U test).

**Figure 4 ijms-23-07949-f004:**
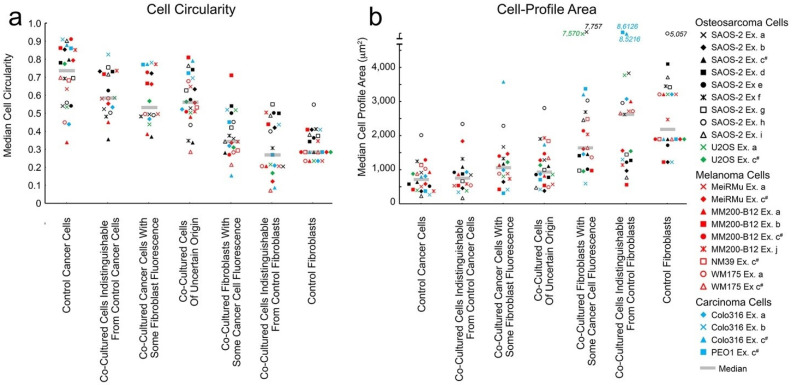
Median cell circularity (**a**) and cell-profile area (**b**) measurements of co-cultured cells grouped according to classification by Cartesian plot, as well as of control cancer cells and fibroblasts, across all 24 co-cultures studied in 10 separate experiments (Ex.). While median values for individual co-cultures are shown, medians across all experiments are also marked within each group by horizontal transparent grey lines. Symbols for co-cultures with SAOS-2 are in black, to aid visual separation from remaining co-cultures with other cancer cell lines. Fibroblasts were labelled with DiD in all experiments. Cancer cells were labelled with DiO, with exception of experiment ‘i’ where SAOS-2 expressed GFP, and also in experiment ‘c’, which is marked c^#^ in the legend and where the orientation of fluorescent labels was swapped. Statistical comparison between all groups shown are provided in Appendix A in the Appendix A. (**a**) Control cancer cells cultured in isolation, had higher cell circularity compared with similarly isolated control fibroblasts (*p* < 0.0001, Wilcoxon signed-rank test). This difference was retained in co-culture, as determined by the comparison of co-cultured cells with fluorescence indistinguishable from either control cancer cells, or control fibroblasts (*p* < 0.0001, Wilcoxon signed-rank test). While there was no statistically significant difference in cell circularity between control cancer cells and co-cultured cells with fluorescence indistinguishable from control cancer cells, co-cultured cancer cells with some fibroblast fluorescence were less circular compared with control cancer cells (*p* < 0.001, Wilcoxon signed-rank test). Co-cultured cancer cells with some fibroblast fluorescence did have generally lower cell circularity compared with co-cultured cells indistinguishable from control cancer cells, but this did not reach statistical significance. Control fibroblasts cultured in isolation did not differ significantly in cell-circularity compared with co-cultured cells with fluorescence indistinguishable from control fibroblasts, but co-cultured cells with fluorescence indistinguishable from control fibroblasts were less circular compared with co-cultured fibroblasts with some cancer cell fluorescence (*p* < 0.03, Wilcoxon signed-rank test). Co-cultured cells of uncertain origin were more circular compared with co-cultured cells identified as fibroblasts either with or without fluorescence from cancer cells, as well as compared against control fibroblast cultures (*p* < 0.0001); (**b**) control cancer cells cultured in isolation, had lower cell-profile area compared with similarly isolated control fibroblasts (*p* < 0.0001, Wilcoxon signed-rank test). This difference was retained in co-culture, as determined by the comparison of co-cultured cells with fluorescence indistinguishable from either control cancer cells, or control fibroblasts (*p* < 0.0001, Wilcoxon signed-rank test). While there was no statistically significant difference in cell-profile area between control cancer cells and co-cultured cells with fluorescence indistinguishable from control cancer cells, co-cultured cancer cells with some fibroblast fluorescence had higher cell-profile area compared with both control cancer cells (*p* < 0.0001, Wilcoxon signed-rank test) and co-cultured cells with fluorescence indistinguishable from control cancer cells (*p* < 0.042, Wilcoxon signed-rank test). There was no statistically significant difference in cell-profile area between: control fibroblasts; co-cultured cells with fluorescence indistinguishable from that of control fibroblasts; and co-cultured fibroblasts with some cancer cell fluorescence. Co-cultured cells of uncertain origin had higher median cell-profile area compared with control cancer cells (*p* < 0.0003, Wilcoxon signed-rank test), and lower median cell-profile area compared with: co-cultured fibroblasts with some fibroblast labelling; co-cultured cells with fluorescence indistinguishable from control fibroblasts; and with control fibroblasts (*p* < 0.0001, Wilcoxon signed-rank test).

**Figure 5 ijms-23-07949-f005:**
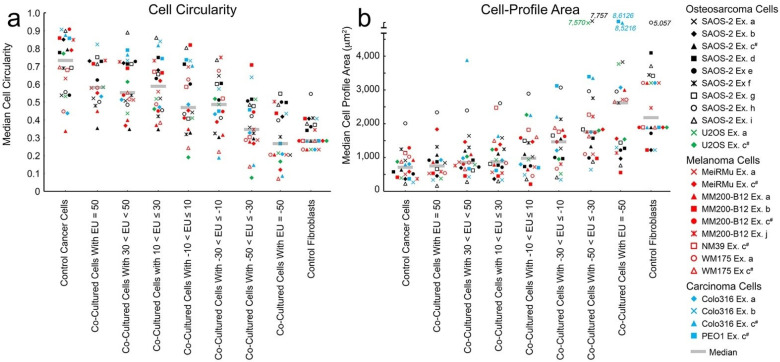
Median cell circularity measurements (**a**) and cell-profile area (**b**) of co-cultured cells grouped according to Exchange Units (EU) in Cartesian plots, as well as of control cancer cells and fibroblasts, across all 24 co-cultures studied in 10 separate experiments (Ex.). While median values for individual co-cultures are shown, medians across all experiments are also marked within each group by horizontal transparent grey lines. Symbols for co-cultures with SAOS-2 are in black, to aid visual separation from remaining co-cultures with other cancer cell lines. Fibroblasts were labelled with DiD in all experiments. Cancer cells were labelled with DiO, with the exception of experiment ‘i’ where SAOS-2 expressed GFP, and also in experiment ‘c’, which is marked c^#^ in the legend and where the orientation of fluorescent labels was swapped. Statistical comparison between all groups shown are provided in Appendix A in Appendix A. (**a**) Control cancer cells were more circular compared with co-cultured cells from 30 < EU ≤ 50 (*p* < 0.003, Wilcoxon signed-rank test), through to EU = 50 (*p* < 0.0001, Wilcoxon signed-rank test). While there was no statistically significant difference in circularity between EU = 50 cells and co-cultured cells in neighbouring groups of cells (30 < EU < 50; 10 < EU ≤ 30; −10 < EU ≤ 10), EU = 50 cells were more circular compared with groups with EU < 10 (*p* < 0.016, Wilcoxon signed-rank test). Co-cultured cells in the group 30 < EU < 50 were more circular, compared with all groups of co-cultured cells with EU ≤ 10 (*p* < 0.05, Wilcoxon signed-tank test). Control fibroblasts were less circular in a statistically significant manner compared with all co-cultured groups of cells (*p* < 0.05), other than compared with cells of EU = −50. EU = −50 cells were less circular compared with all other groups of co-cultured cells (*p* < 0.009, Wilcoxon signed-rank test). Co-cultured cells in the group −50 < EU ≤ −30, were less circular compared with all groups with EU > −30 (*p* < 0.015, Wilcoxon signed-rank test); (**b**) control cancer cells had smaller median cell-profile areas compared with all groups of co-cultured cells ranging from 30 < EU < 50 (*p* < 0.01, Wilcoxon signed-rank test), through to EU = 50 (*p* < 0.0001, Wilcoxon signed-rank test). The co-cultured cell group with EU = 50 had lower cell-profile area compared with all other groups of co-cultured cells, and this reached statistical significance for 30 < EU < 50 (*p* < 0.035, Wilcoxon signed-rank test), and for groups where EU ≤ −10 (*p* < 0.003, Wilcoxon signed-rank test). Co-cultured cells in the group 30 < EU < 50 had lower cell-profile area compared with all groups with EU ≤ 30, but this only reached statistical significance where EU ≤ −10 (*p* < 0.0001, Wilcoxon signed-rank test). Control fibroblasts and EU = −50 cells had statistically significant greater cell-profile area compared with all groups of co-cultured cells with EU > −30 (*p* < 0.0002, Wilcoxon signed-rank test). Co-cultured cells in the group −50 < EU ≤ −30, had larger cell-profile area compared with all groups with EU > −30 (*p* < 0.0004, Wilcoxon signed-rank test).

**Figure 6 ijms-23-07949-f006:**
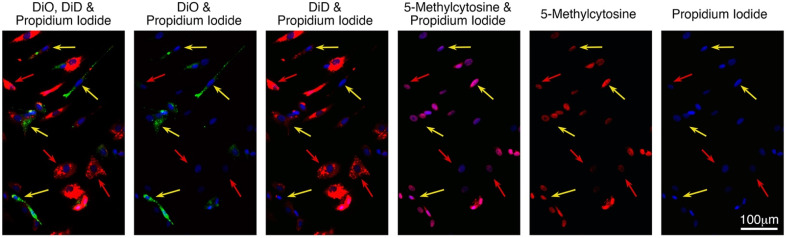
Photomicrographs from experiment ‘p’ of fibroblasts (pre-labelled with DiD) co-cultured for 24 h with SAOS-2 (pre-labelled with DiO), with co-cultures then further labelled with propidium iodide for DNA and by fluorescence immunohistochemistry for 5mc. Channels for DiO, DiD, 5mc and propidium iodide are shown. DiD and DiO seen previously (Figure 1), with fibroblasts (red arrows) distinguishable from SAOS-2 (yellow arrows), which in this particular visual field all received at least some fluorescent contents from fibroblasts. While the difference in propidium iodide fluorescence between fibroblasts and SAOS-2 seen in control cultures appeared maintained (Appendix A), co-cultured fibroblasts seemed to have less intense 5mc fluorescence compared with fibroblasts cultured in isolation (Appendix A).

**Figure 7 ijms-23-07949-f007:**
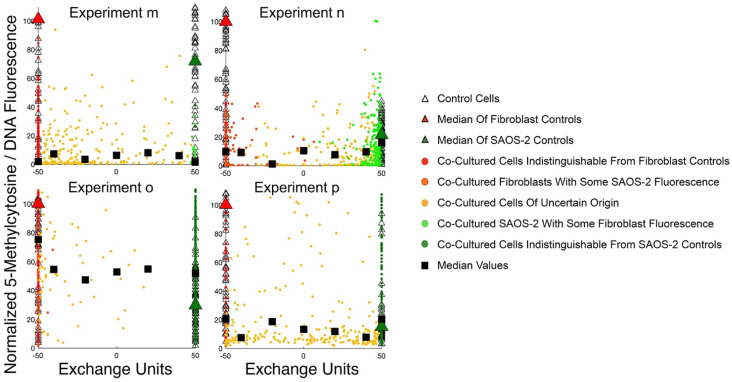
Cartesian plot analysis results for normalised 5mc for all experiments ‘m’ to ‘p’. ‘Red Cells’ comprise fibroblasts in these plots, while ‘Green Cells’ are SAOS-2. Control fibroblasts cultured in isolation (medians shown by large red triangles) had higher normalised 5mc compared with control SAOS-2 (medians shown by large green triangles) (*p* < 0.035, paired t-test considering all experiments together; *p* < 0.015, Mann–Whitney U tests within individual experiments). Co-cultured fibroblasts with fluorescence indistinguishable from controls (red circles) had generally lower normalised 5mc compared with control fibroblasts, (black squares at EU = −50 are medians of co-cultures; *p* < 0.25, paired t-test considering all experiments together; *p* < 0.0025, Mann–Whitney U tests within individual experiments). There was no clear relationship between normalised 5mc in co-cultured cells and distribution according to Cartesian plot.

**Table 1 ijms-23-07949-t001:** The number of co-cultured cells according to Cartesian plot grouping, as well as of control cells cultured in isolation, in each experiment. The relative percentage of cells in each co-cultured group is indicated in brackets (%) to aid comparison. Ostesoarcoma (SAOS-2, U2OS), melanoma (MeIRMu, MM200-B12, NM39, WM175), colon carcinoma (Colo316) and ovarian carcinoma (PEO1) cells were studied. All experiments were with human dermal fibroblasts, labelling fibroblasts with DiD and cancer cell lines with DiO, except for experiment ‘i’ where SAOS-2 expressed GFP, and experiment ‘c’ (marked *), where human gingival fibroblasts were used and labelling had the opposite orientation, such that fibroblasts were pre-labelled with DiO and cancer cells with DiD.

			Co-Cultured Cells According to Cartesian Plot Grouping						
Cell Line	Exp.	Control Fibroblasts	Cells Indistinguishable from Fibroblast Controls	Fibroblasts with Some Cancer Cell Labelling	Cells of Uncertain Origin	Cancer Cells with Some Fibroblast Labelling	Cells Indist-inguishable from Cancer Cell Controls	Total Co-Cultured Cells	Control Cancer Cells
SAOS-2	a	100	103 (43.1%)	1 (0.4%)	52 (21.8%)	81 (33.9%)	2 (0.8%)	239	100
SAOS-2	b	96	33 (28.0%)	0 (0%)	2 (1.7%)	41 (34.7%)	42 (35.6%)	118	40
SAOS-2	c*	102	0 (0%)	10 (13.3%)	30 (40.0%)	17 (22.7%)	18 (23.7%)	75	97
SAOS-2	d	100	464 (50.0%)	199 (21.4%)	123 (13.3%)	0 (0%)	142 (15.3%)	928	97
SAOS-2	e	86	212 (21.2%)	416 (41.6%)	365 (36.5%)	0 (0%)	6 (0.6%)	999	95
SAOS-2	f	97	197 (20.4%)	40 (4.1%)	50 (5.2%)	178 (18.4%)	501 (51.9%)	966	100
SAOS-2	g	97	184 (24.9%)	5 (0.7%)	206 (27.8%)	0 (0%)	345 (46.6%)	740	98
SAOS-2	h	99	73 (7.4%)	54 (5.5%)	423 (43.1%)	223 (22.7%)	209 (21.3%)	982	92
SAOS-2	i	98	406 (46.6%)	38 (4.4%)	426 (48.9%)	0 (0%)	2 (0.2%)	872	99
U2OS	a	100	162 (48.8%)	3 (0.9%)	56 (16.9%)	102 (30.7%)	9 (2.7%)	332	98
U2OS	c*	102	6 (5.9%)	1 (1.0%)	62 (61.4%)	32 (31.7%)	0 (0%)	101	98
MeIRMu	a	100	168 (48.1%)	0 (0%)	19 (5.4%)	128 (36.7%)	34 (9.7%)	349	96
MeIRMu	c*	94	5 (6.9%)	1 (1.6%)	11 (17.2%)	43 (67.2%)	5 (5.6%)	64	102
MM200-B12	a	100	159 (47.3%)	0 (0%)	40 (11.9%)	119 (35.4%)	18 (5.4%)	336	98
MM200-B12	b	96	79 (69.3%)	3 (2.6%)	5 (4.4%)	24 (21.1%)	3 (2.6%)	114	68
MM200-B12	c*	102	0 (0%)	16 (27.6%)	40 (69.0%)	2 (3.4%)	0 (0%)	58	100
MM200-B12	j	98	44 (21.2%)	57 (27.4%)	21 (10.1%)	20 (9.6%)	66 (31.7%)	208	100
NM39	c*	102	0 (0%)	9 (11.5%)	69 (88.5%)	0 (0%)	0 (0%)	78	88
WM175	a	100	168 (49.4%)	2 (0.6%)	17 (5.0%)	122 (35.9%)	31 (9.1%)	340	97
WM175	c*	102	3 (4.9%)	21 (35.6%)	35 (59.3%)	0 (0%)	0 (0%)	59	94
Colo316	a	100	188 (48.0%)	3 (0.8%)	79 (20.2%)	113 (28.8%)	9 (2.3%)	392	100
Colo316	b	96	107 (54.0%)	3 (1.5%)	24 (12.2%)	34 (17.3%)	28 (14.1%)	196	75
Colo316	c*	102	6 (5.8%)	50 (49.5%)	34 (33.7%)	11 (10.9%)	0 (0%)	101	89
PEO1	c*	102	2 (2.5%)	41 (50.6%)	35 (43.2%)	3 (3.7%)	0 (0%)	81	92

There was appreciable uptake of fibroblast label by cancer cells across experiments, such that in some experiments no or very few cancer cells indistinguishable from control cancer cells were seen (experiments: ‘a’, ‘e’ and ‘i’ with SAOS-2; ‘a’ and ‘c’ with U2OS; ‘b’ and ‘c’ with MM200-B12; ‘c’ with NM39; ‘a’ with Colo316; and ‘c’ for WM175, Colo316 and PEO1 cells). Cells in the group of ‘cancer cells with some fibroblast labelling’ were seen in 18 of 24 co-cultures, comprising from 3.4% to 67.2% of all co-cultured cells. Co-cultured cells of uncertain and in origin were present in all experiments, comprising 1.7% (SAOS-2, experiment ‘b’) to 88.5% (NM39, experiment ‘c’) of cells characterised. Fibroblasts with some cancer cell labelling were prominent in some experiments (experiments ‘d’ and ‘e’ with SAOS-2, experiments ‘c’ and ‘j’ for MM200-B12; and for WM175, Colo316 and PEO-1 cells in experiment ‘c’), but otherwise comprised a small proportion of cells (SAOS-2 experiments ‘a’, ‘b’, ‘c’, ‘f’, ‘g’, ‘i’; U2OS experiments ‘a’ and ‘c’; MeIRMu experiments ‘a’ and ‘c’; MM200-B12 experiments ‘a’, ‘b’; NM39 experiment ‘c’; WM175 experiment ‘a’; Colo316 experiment ‘b’). Cells indistinguishable from fibroblasts cultured in isolation were prominent across experiments, with the exception of experiment ‘c’ where fibroblasts had much higher uptake of fluorescence from cancer cells compared with other experiments.

**Table 2 ijms-23-07949-t002:** The number of co-cultured cells according to Exchange Units (EU) in each experiment. The relative percentage of cells in each co-cultured group is indicated in brackets (%) to aid comparison. Ostesoarcoma (SAOS-2, U2OS), melanoma (MeIRMu, MM200-B12, NM39, WM175), colon carcinoma (Colo316) and ovarian carcinoma (PEO1) cells were studied. All experiments were with human dermal fibroblasts, labelling fibroblasts with DiD and cancer cell lines with DiO, except for experiment ‘i’ where SAOS-2 expressed GFP, and experiment ‘c’ (marked *), where human gingival fibroblasts were used and labelling had the opposite orientation, such that fibroblasts were pre-labelled with DiO and cancer cells with DiD. To aid comparison of results, EU for experiment ‘c’ were multiplied by −1, effectively inverting the order of groups to bring them into alignment with other experiments.

		Co-Cultured Cells Grouped According to Exchange Units (EU)						
Cell Line	Exp.	EU = −50	−50 < EU ≤ −30	−30 < EU ≤ −10	−10 < EU ≤ 10	10 < EU ≤ 30	30 < EU < 50	EU = 50
SAOS-2	a	103 (43.1%)	1 (0.4%)	4 (1.7%)	17 (7.1%)	29 (12.1%)	83 (34.7%)	2 (0.8%)
SAOS-2	b	33 (28.0%)	0 (0%)	0 (0%)	0 (0%)	6 (5.1%)	37 (31.4%)	42 (35.6%)
SAOS-2	c*	0 (0%)	2 (2.6%)	7 (9.2%)	10 (13.2%)	12 (15.8%)	27 (35.5%)	18 (23.7%)
SAOS-2	d	464 (50.0%)	142 (15.3%)	56 (6.0%)	60 (6.5%)	63 (6.8%)	1 (0.1%)	142 (15.3%)
SAOS-2	e	212 (21.2%)	531 (53.2%)	97 (9.7%)	86 (8.6%)	52 (5.2%)	15 (1.5%)	6 (0.6%)
SAOS-2	f	197 (20.4%)	51 (5.2%)	18 (1.8%)	22 (2.3%)	31 3.2(%)	153 (15.7%)	501 (51.9%)
SAOS-2	g	184 (24.9%)	49 (6.6%)	27 (3.6%)	15 (2.0%)	28 (3.8%)	92 (12.4%)	345 (46.6%)
SAOS-2	h	73 (7.4%)	133 (13.5%)	85 (8.7%)	92 (9.4%)	114 (11.6%)	276 (28.1%)	209 (21.3%)
SAOS-2	i	406 (46.6%)	178 (20.4%)	110 (12.6%)	87 (10.0%)	63 (7.2%)	26 (3.0%)	2 (0.2%)
U2OS	a	162 (48.8%)	3 (0.9%)	6 (1.8%)	12 (3.6%)	36 (10.8%)	104 (31.3%)	9 (2.7%)
U2OS	c*	6 (5.9%)	1 (1.0%)	5 (5.0%)	8 (7.9%)	21 (20.8%)	60 (59.4%)	0 (0%)
MeIRMu	a	168 (48.1%)	2 (0.6%)	0 (0%)	4 (1.1%)	12 (3.4%)	129 (37.0%)	34 (9.7%)
MeIRMu	c*	5 (6.9%)	18 (25.0%)	23 (31.9%)	12 (16.7%)	7 (9.7%)	3 (4.2%)	5 (5.6%)
MM200-B12	a	159 (47.3%)	2 (0.6%)	4 (1.2%)	4 1.2(%)	20 (6.0%)	129 (38.4%)	18 (5.4%)
MM200-B12	b	79 (69.3%)	3 (2.6%)	0 (0%)	2 (1.8%)	6 (5.3%)	21 (18.4%)	3 (2.6%)
MM200-B12	c*	0 (0%)	20 (33.9%)	22 (37.3%)	9 (15.3%)	5 (8.5%)	3 (5.1%)	0 (0%)
MM200-B12	j	44 (21.2%)	69 (33.2%)	5 (2.4%)	0 (0%)	1 (0.5%)	23 (11.1%)	66 (31.7%)
NM39	c*	0 (0%)	12 (15.4%)	36 (46.2%)	28 (35.9%)	2 (2.6%)	0 (0%)	0 (0%)
WM175	a	168 (49.4%)	2 (0.6%)	3 (0.9%)	5 (1.5%)	13 (3.8%)	118 (34.7%)	31 (9.1%)
WM175	c*	3 (4.9%)	7 (11.5%)	8 (13.1%)	21 (34.4%)	17 (27.9%)	5 (8.2%)	0 (0%)
Colo316	a	188 (48.0%)	2 0.5(%)	8 (2.0%)	10 (2.6%)	29 (7.4%)	146 (37.2%)	9 (2.3%)
Colo316	b	107 (54.0%)	9 (4.5%)	9 4.5(%)	4 (2.0%)	14 (7.1%)	27 (13.6%)	28 (14.1%)
Colo316	c*	6 (5.8%)	43 (41.7%)	14 (13.6%)	8 (7.8%)	19 (18.4%)	13 (12.6%)	0 (0%)
PEO1	c*	2 (2.5%)	22 (26.8%)	23 (28%)	23 (28%)	10 (12.2%)	2 (2.4%)	0 (0%)

In nine co-cultures, there was a preponderant distribution of cells with −50 ≤ EU < −10, where over 60% of cells with a more ‘fibroblast-like’ pattern of fluorescence (SAOS-2 experiments ‘d’, ‘e’, ‘i’; MeIRMu experiment ‘c’; MM200-B12 experiments ‘b’, ‘c’; NM39 experiment ‘c’; Colo316 experiments ‘b’, ‘c’), while in experiment ‘c’ with PEO1, over 56% of cells were more similar to fibroblasts as opposed to only14.6% of cells that were more similar to cancer cells in fluorescence. Co-cultures where over 60% of cells had a more ‘cancer cell-like’ pattern of fluorescence from 10 < EU ≤ 50, was seen in six experiments (SAOS-2 in experiments ‘b’, ‘c’, ‘f’, ‘g’, ‘h’; U2OS experiment ‘c’). ‘Fibroblast-like’ (−50 ≤ EU < −10) and ‘cancer cell-like’ cells (10 < EU ≤ 50) each comprised close to 50% of cells in seven co-cultures (SAOS-2 experiment ‘a’; U2OS experiment ‘a’; MeIRMu experiment ‘a’; MM200-B12 experiments ‘a’, ‘j’; WM175 experiment ‘a’; Colo316 experiment ‘a’). In most of these co-cultures, comparatively few cells were in the mid-range of −10 < EU ≤ 10, but in one co-culture, both ‘cancer cell-like’ and ‘fibroblast-like’ cells comprised less than 40% of co-cultured cells, and over 34% of cells were in this mid domain. Generally speaking, all co-cultures had notable paucity of cells across several of the defined EU groupings, so that with exception of one co-culture (SAOS-2 experiment ‘h’), up to four of the EU domains defined above in each co-culture, had fewer than 5% of co-cultured cells (one domain in SAOS-2 experiment ‘d’; MeIRMu experiment ‘c’; Colo316 experiment ‘c’); two domains in eight co-cultures (SAOS-2 experiments ‘c’, ‘e’, ‘f’, ‘i’; U2OS experiment ‘c’; MM200-B12 experiments ‘c’, ‘j’; WM175 experiment ‘c’); three domains in six co-cultures (SAOS-2 experiments ‘a’, ‘b’, ‘g’; MMB-12 experiment ‘a’; Colo316 experiment ‘b’; PEO1 experiment ‘c’); and four domains in six co-cultures (U2OS experiment ‘a’; MeIRMu experiment ‘a’; MM200-B12 experiment ‘b’; NM39 experiment ‘c’; WM175 experiment ‘a’; Colo316 experiment ‘a’).

## Data Availability

An Excel spreadsheet containing all input data for Cartesian plot analysis is provided in Appendix A.

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
