# Peer review of "A Novel Cartesian Plot Analysis for Fixed Monolayers That Relates Cell Phenotype to Transfer of Contents between Fibroblasts and Cancer Cells by Cell-Projection Pumping"

_ijms, 2022, doi:10.3390/ijms23147949_

Round 1

Reviewer 1 Report

The authors presented study entitled “A Novel Cartesian Plot Analysis for Fixed Monolayers That Relates Cell Phenotype to Transfer of Contents Between Fibroblasts and Cancer Cells by Cell-Projection Pumping”.

This is a very interesting study aimed at elucidating the mechanism of cytoplasm transfer between cultured human fibroblasts and cancer cells using fluorescent label.

In particular, this study represents an evolution of a previous one showing that the uptake of fibroblast content by tumor cells increases their proliferation, migration, size and complexity, however a quantitative method for identifying subpopulations of co-cultured cells and transfer from control populations was lacking.

In the present study the authors overcome the limitation of previous studies by describing a new Cartesian plot image analysis that quantifies fluorescent label transfers between co-cultured cell populations observed in fixed monolayers.

The study is well conducted and the results are presented and discussed in a clear and detailed manner. This reviewer agrees with the authors on the important impact of the study in evaluating the relationship between cell projection pumping and possible phenotypic effects and for 5mc quantification, as it is a faster analytical method than the other single-cell analytical method, showing results consistent with the microscopic observations previously reported.

The introduction section provide sufficient background however, in view of the results obtained, this reviewer suggests describing cancer-associated fibroblasts: function and main reasons why these cell populations are extensively studied to design anticancer drugs.

Please check for a possible typo between lines 766 and 767.

Author Response

Thank you for your thoughtful review and suggestions. We have responded to all of the Reviewer's comments, and believe that the paper is much improved as a result.

All changes in the manuscript are described in detail in the response to Reviewers, and text changes are also shown in the 'track changes' document'.  Please note that the final revised document with 'track changes accepted', has text shuffled into different positions relative to the images and tables, and these are not recorded in the 'track-changes document', because doing so would have obscured the more important text changes and made them difficult for reviewers to find.  The changes in image position were necessary to accommodate the increased size of Figure 1 as requested by Reviewer 2, and also to sensibly locate images and tables relative to the modified text.

Whilst responding to the Reviewers comments, we noticed some typos and minor mistakes in section numbering, and these have been corrected as marked in the 'track-changes document'.

REVIEWER 1, POINT 1: 

The introduction section provide sufficient background however, in view of the results obtained, this reviewer suggests describing cancer-associated fibroblasts: function and main reasons why these cell populations are extensively studied to design anticancer drugs.

RESPONSE TO REVIEWER 1, POINT 1:

This is a very sensible suggestion, and we have included the following new text in the introduction, and also slightly modified the original relevant sentence, to better describe the significance of cancer associated fibroblasts.

Track Changes Document Page 3, Lines 108 to 116: 'Cancer associated fibroblasts are an important component of the tumor microenvironment, and influence responsiveness to cancer treatment via a range of mechanisms including their: effect on cancer cell stemness [40]; increasing epithelial to mesenchymal transformation via Transforming Growth Factor-beta release [41]; modifying chemotherapy responsiveness [42]; and aiding immune evasion through via secretory products and evocation of a dense fibrotic stroma that obstructs lymphocyte migration [43-45]. Because of this, cancer associated fibroblasts have been identified as a target for therapeutic intervention [46]. It is of interest for the current work that cancer associated fibroblasts have hypomethylated DNA [27, 47-50].' 

REVIEWER 1, POINT 2: 

Please check for a possible typo between lines 766 and 767.

RESPONSE TO REVIEWER 2, POINT 2:

We have checked the lines indicated, and the only possible typo we were able to find was an additional and unnecessary space inserted between the words 'PBS)' and 'for'.  That is now removed,  and due to other changes in the text, that correction is now on line 795 of the 'track changes document'.  We do hope that this addresses the Reviewer's concern, but will be happy to make any necessary emendation if there is an expression we have not noticed that introduces unhelpful ambiguity or confusion. 

Reviewer 2 Report

Mahadevan and colleagues present in this manuscript a methodological approach to determine transfer of content between different cell lines in co-culture. This method is designed to study cross-talk between cancer and stromal cells and has potential further application in the microenvironmental studies. The authors also tested the impact of the transfer of cellular content between cell lines by determining the changes in DNA methylation. In summary, the proposed approach sounds interesting and may potentially have broad application in co-culture studies.

 Specific concerns:

1.       Fig.1, A higher magnification and the outline of the cell profile will help reader capture the overlap of two colors in the same cell.

2.       The column title in Table 1 is hard to read. Please reformat it.

3.       Does the transfer of fluorescence between cells require cell-to-cell contact? Is exosome also able to deliver the lipophilic fluorescent dyes? If that is the case, how do you know the transfer of content is driven by exosome or cell-projection pumping?

4.       How could the 5mc in-situ labeling (genomic DNA methylation) reflect the differentiation of CAF? Could you measure some CAF markers (eg: SMA) by FACS or IF staining, and correlate it to the transfer of fluorescent dye? This could inform that whether the transfer of membrane content (or cytoplasmic content) could actually have biological impact on the receipt cells.

Author Response

Thank you for your thoughtful review and suggestions. We have responded to all of the Reviewer's comments, and believe that the paper is much improved as a result.

All changes in the manuscript are described in detail in the response to Reviewers, and text changes are also shown in the 'track changes' document'.  Please note that the final revised document with 'track changes accepted', has text shuffled into different positions relative to the images and tables, and these are not recorded in the 'track-changes document', because doing so would have obscured the more important text changes and made them difficult for reviewers to find.  The changes in image position were necessary to accommodate the increased size of Figure 1 as requested by Reviewer 2, and also to sensibly locate images and tables relative to the modified text.

Whilst responding to the Reviewers comments, we noticed some typos and minor mistakes in section numbering, and these have been corrected as marked in the 'track-changes document'.

REVIEWER 2, POINT 1:

Fig.1, A higher magnification and the outline of the cell profile will help reader capture the overlap of two colors in the same cell.

RESPONSE TO REVIEWR 2, POINT 1:

Inspecting the image, we are glad that the Reviewer noticed the difficulty and agree that cells in the image in the first submission, were too small to easily see dual labelling.  We have substantially increased the size of this figure on the page and this greatly increases the surface area of each cell visible. We believe it is now much easier for the reader to properly see the dual labelling.  

REVIEWER 2, POINT 2:

The column title in Table 1 is hard to read. Please reformat it.

RESPONSE TO REVIEWER 2, POINT 2:

The font size of text of the column titles has been increased to improve readability.

REVIEWER 2, POINT 3:

Does the transfer of fluorescence between cells require cell-to-cell contact? Is exosome also able to deliver the lipophilic fluorescent dyes? If that is the case, how do you know the transfer of content is driven by exosome or cell-projection pumping?

RESPONSE TO REVIEWER 2, POINT 3:

The reviewer raises an important point. Exosomes can and do transfer membrane content between cells, but the quantity of transfer is much less compared with the transfers we observe in our system. We have confirmed this in experiments where conditioned medium containing any shed exosomes fails to transfer appreciable fluorescence to the opposing cultured cells.  Also, our time-lapse recordings  show that bulk fluorescence transfer we study is related to sporadic occurrences of cell-projection pumping, and is not slow, consistent and uniform as would be expected were exosomes responsible. The same time-lapse recordings exclude significant transfer via tunnelling nanotubes.   Finally, much of our data is in SAOS-2, where we have found cell-projection pumping to be the predominant mechanism whereby transfers occur.  We clarify this in the manuscript by insertion of the following new text:

Track Changes Document Pages 1 to 2, Lines 45 to 49: 'Notably, a major role for either exosomes or tunnelling nanotubes is excluded in our cultures by video microscopy demonstrating highly localized, discrete and rapid transfer events independent of tunneling nanotubes and inconsistent with exosomes, as well as in experiments with conditioned media containing putative exosomes [1-4], so that cell-projection pumping is responsible for the transfers we study.'

REVIEWER 2, POINT 4:

How could the 5mc in-situ labeling (genomic DNA methylation) reflect the differentiation of CAF? Could you measure some CAF markers (eg: SMA) by FACS or IF staining, and correlate it to the transfer of fluorescent dye? This could inform that whether the transfer of membrane content (or cytoplasmic content) could actually have biological impact on the receipt cells.

RESPONSE TO REVIEWER 2, POINT 4:

Reviewer 2 is right to point out that we have not specifically explored markers of CAF differentiation, and that the current data does not support a role for cell-projection pumping in CAF differentiation.  While we did try to make clear in the original manuscript that we do not think cell-projection pumping plays such a role, we also infered that the co-culture system might generate CAF-like cells, probably via secreted cancer cell products and or exosomes. 

Although we do still think it is reasonable to suggest the possibility that CAFs or at least CAF-like cells might be generated by the co-culture conditions we study, we also accept the point that this is only a possibility that would require further study to establish.  We have added the following text to address this in the manuscript:  

Track Changes Document Pages 15 to 16, Lines 542 to 557 (including marked deleted text):  'We have not characterized the fibroblasts in our co-cultures to determine if their phenotype is in alignment with that of cancer associated fibroblasts. However, the lower normalized 5mc of co-cultured fibroblasts is consistent with separate literature where cancer associated fibroblasts are reported as hypomethylated compared with normal fibroblasts [47, 48].  This finding in the current work, suggests that the co-culture system used may provide a good model for generating artificial cancer associated fibroblasts for separate in-vitro study, but further work beyond the scope of the current investigation would be required to determine if this is the case. Irrespective, consistency of lower normalized 5mc in co-cultured fibroblasts with low 5mc found in cancer associated fibroblasts, does lend further confidence in the utility of the Cartesian plot method.'

Regarding the question Reviewer 2 asks if we could correlate CAF differentiation with transfer of fluorescent dye, we did not see any correlation between normalized 5mC and fluorescence transfer, and so have no evidence for this.  However, it might be interesting to explore in further work, if fluorescence transfer correlates with expression of CAF phenotype, such as SMA expression.   

This is addressed in the manuscript where the text from the original manuscript reads:

Track Changes Document Page 16, Lines 559 to 568:  'It has been reasonable to speculate that cell-projection pumping might play a role in development of cancer associated fibroblasts, especially since some transfer from SAOS-2 to fibroblasts has been reported in earlier work from this laboratory [1, 2]. However, despite seeing morphological changes in co-cultured SAOS-2 consistent with cell-projection pumping, there was no similar correlation between normalized 5mc in either SAOS-2 or HDF, and transfer of fluorescence between the two cell populations. From this, the possibility that cell-projection pumping drives cancer associated fibroblast differentiation is in large part refuted by the current results. In light of the current data, it seems more likely that cancer associated fibroblast differentiation is driven by soluble factors and or exosomes released by cancer cells, and this is consistent with the literature [62].'

And the following new passage has been added

Track Changes Document Page 16, Lines 568 to 572:  'Nonetheless, the possibility that cell-projection pumping might contribute to cancer associated fibroblast differentiation, could be further studied using the Cartesian plot method to correlate expression of cancer associated fibroblast markers such as smooth muscle cell actin, with fluorescence uptake from cancer cells.'